# Current Status of Herbicide Resistance in the Iberian Peninsula: Future Trends and Challenges

Joel Torra [1,*], José M. Montull [1], Isabel M. Calha [2], María D. Osuna [3], Joao Portugal [4,5] and Rafael de Prado [6]

1. Department d'Hortofruticultura, Botànica i Jardineria, Agrotecnio-CERCA Center, Universitat de Lleida, 25198 Lleida, Spain; josemaria.montull@udl.cat
2. Instituto Nacional de Investigação Agrária e Veterinária (INIAV, IP), Quinta do Marquês, 2784-505 Oeiras, Portugal; isabel.calha@iniav.pt
3. Plant Protection Department, Extremadura Scientific and Technological Research Center (CICYTEX), Ctra. de AV, km 372, 06187 Guadajira, Spain; mariadolores.osuna@juntaex.es
4. Biosciences Department, Polytechnic Institute of Beja, 7800-295 Beja, Portugal; jportugal@ipbeja.pt
5. VALORIZA-Research Centre for Endogenous Resource Valorization, Polytechnic Institute of Portalegre, 7300-555 Portalegre, Portugal
6. Department of Agricultural Chemistry, Edaphology and Microbiology, University of Cordoba, 14014 Córdoba, Spain; qe1pramr@uco.es
* Correspondence: joel.torra@udl.cat

**Abstract:** The evolution of herbicide resistance in weeds has emerged as one of the most serious threats to sustainable food production systems, which necessitates the evaluation of herbicides to determine their efficacy. The first herbicide resistance case in the Iberian Peninsula was reported about 50 years ago, wherein *Panicum dichotomiflorum* was found to be resistant (R) to atrazine in Spanish maize fields. Since then, herbicide resistance has evolved in 33 weed species, representing a total of 77 single-herbicide-resistance cases in this geographic area: 66 in Spain and 11 in Portugal. Changes in agricultural practices, namely the adoption of non-tillage systems and the increased use of herbicides, led to the selection of weed biotypes resistant to a wide range of herbicides. Nowadays the most important crops in Spain and Portugal (maize, winter cereals, rice, citrus, fruits, and olive orchards) are affected, with biotypes resistant to several mechanisms of action (MoAs), namely: ALS inhibitors (20 species), ACCase inhibitors (8 species), PS II inhibitors (18 species), and synthetic auxin herbicides (3 species). More recently, the fast increase in cases of resistance to the EPSPS-inhibiting herbicide glyphosate has been remarkable, with 11 species already having evolved resistance in the last 10 years in the Iberian Peninsula. The diversity of resistance mechanisms, both target-site and non-target-site, are responsible for the resistance to different MoAs, involving point mutations in the target site and enhanced rates of herbicide detoxification, respectively. More serious are the 13 cases reported with multiple-herbicide resistance, with three cases of resistance to three–four MoAs, and one case of resistance to five MoAs. Future research perspectives should further study the relationship between management strategies and the occurrence of TSR and NTSR resistance, to improve their design, develop monitoring and diagnostic tools for herbicide resistance, and deepen the study of NTSR resistance.

**Keywords:** *Amaranthus palmeri*; enhanced metabolism; herbicide resistance cases; *Lolium* spp.; multiple-herbicide resistance; point mutations; Portugal; Spain

## 1. Introduction

Herbicides still play a pivotal role in crop protection for weed management worldwide [1]. Therefore, it is well known that high herbicide pressure selects for evolved resistance in weed populations [2]. Herbicide resistance is defined as the acquired inheritable trait of weeds to survive and reproduce under herbicide exposure. The genetic variability and reproductive biology of weeds are the most important drivers of resistance

evolution [3]. Unfortunately, the Iberian Peninsula (close to 600,000 km$^2$), which comprises Portugal and Spain in the south-western corner of Europe, is not an exception. Spain, which represents more than 80% of the area covered in this review, ranks 6th in the world, and 2nd in Europe (after France) in terms of unique herbicide-resistant (HR) weed cases [4]. According to the updated information that will be provided in this review, it ranks 5th, only behind the biggest countries in the world in terms of both food production and surface area: Canada, USA, Brazil, and Australia. This review will provide insights into the factors contributing to this worrying scenario and expected future trends in herbicide resistance development.

At the global level, estimations indicate that weeds cause potential yield losses of around 34% in main crops, high above of those caused by insects (18%) or diseases (16%) [5]. To the best of our knowledge there is no such up-to-date information available for Spain or Portugal, though existing data do indicate that herbicide consumption is increasing and higher compared to other pesticides. For example, Spain is the second European country in terms of herbicide usage [6]. Moreover, around 20,200 tons of herbicides were commercialized in 2020 in Spain, which is 4% more than in 2019 [7]. This herbicide expenditure highlights how weeds can impact the agriculture of the Iberian Peninsula.

The main goal of this paper is to update previous reviews and reports about herbicide resistance in Spain and Portugal. Particularly, its aim is to update the information on the status of resistant weeds both in crop and non-crop lands in the Iberian Peninsula, which was last compiled in 2008 [8]. It also aims to update and provide additional information to that compiled in the International Survey of Herbicide-Resistant Weeds [4], because not all resistance cases are reported there, even if they have been published in peer-reviewed international journals with SCI impact factors. In early 2022, 46 unique resistance cases were reported in Heap's database [4]. The criteria used in this review to compile all the credible resistance cases in Spain and Portugal, besides those already included in the mentioned database, added new cases found thanks to: (a) a bibliographic review of the Web of Science and Scopus citation databases of all HR cases in articles from SCI international peer-reviewed journals; (b) a bibliographic review of the communications presented in all instances of congress of the Spanish Weed Science Society; and (c) those HR cases that can be confirmed thanks to the reviewed research groups of these authors. Most HR cases are already reported in Heap's database [4] or supported by an article in an SCI journal. A total of 31 cases from the Iberian Peninsula are not included in Heap's database [4], 6 for Portugal (Table 1), and 25 for Spain (Table 2). There are different explanations for this fact, one of them being that researchers need to be registered in the database to be able to report the resistant case themselves.

**Table 1.** Herbicide-resistant weeds in Portugal. Last column details the reference acknowledging the resistant case. Resistance cases under grey background (6 in total) are those not included in Heap database [4]. * Personal communication.

| Number | Species | Year | MoAs | Herbicides | Crops/Situations | References |
|---|---|---|---|---|---|---|
| 1 | *Alisma plantago-aquatica* | 1995 | ALS inhibitors (2) | Bensulfuron | Rice | [9,10] |
| 2 | *Chenopodium album* | 2003 | Photosystem II inhibitors (5) | Atrazine | Corn | [8] |
| 3 | *Conyza bonariensis* | 2010 | EPSP synthase inhibitors (9) | Glyphosate | Orchards | [4] |
| 4 | *Conyza canadensis* | 2011 | EPSP synthase inhibitors (9) | Glyphosate | Olive | [4] |
| 5 | *Lolium perenne* | 2013 | EPSP synthase inhibitors (9) | Glyphosate | Vineyards | [11] |
| 6 | *Echinochloa phyllopogon* | 2015 | ALS inhibitors (2) | Penoxsulam | Rice | Calha * |
| 7 | *Echinochloa crus-galli* | 2015 | ALS inhibitors (2) | Penoxsulam | Rice | Calha * |
| 8 | *Echinochloa crus-galli* | 2018 | EPSP synthase inhibitors (9) | Glyphosate | Pomegranates, corn, vineyards | [12] |
| 9 | *Papaver rhoeas* | 2018 | ALS inhibitors (2) | Tribenuron | Cereals | Portugal * |
| 10 | *Echinochloa phyllopogon* | 2020 | ACCase inhibitors (1) | Profoxydim | Rice | Calha * |
| 11 | *Echinochloa phyllopogon* | 2020 | ALS inhibitors (2), ACCase inhibitors (1) | Profoxydim, penoxsulam | Rice | Calha * |

**Table 2.** Herbicide-resistant weeds of Spain. Last column details the reference acknowledging the resistant case. Resistance cases under grey background (25 in total) are those not included in Heap database [4].

| Number | Species | Year | MoAs | Herbicides | Crops/Situations | References |
|---|---|---|---|---|---|---|
| 1 | *Panicum dichotomiflorum* | 1981 | Photosystem II inhibitors (5) | Atrazine | Corn, cropland | [13] |
| 2 | *Amaranthus hybridus* | 1985 | Photosystem II inhibitors (5) | Atrazine | Corn | [14] |
| 3 | *Amaranthus blitoides* | 1986 | Photosystem II inhibitors (5) | Simazine | Orchards | [14] |
| 4 | *Amaranthus retroflexus* | 1986 | Photosystem II inhibitors (5) | Atrazine | Corn | [14] |
| 5 | *Setaria faberi* | 1987 | Photosystem II inhibitors (5) | Atrazine | Corn | [15] |
| 6 | *Conyza canadensis* | 1987 | Photosystem II inhibitors (5) | Atrazine, simazine | Corn, orchards, roadsides | [4,16] |
| 7 | *Setaria viridis* | 1987 | Photosystem II inhibitors (5) | Atrazine | Corn | [15] |
| 8 | *Setaria pumila* | 1987 | Photosystem II inhibitors (5) | Atrazine | Corn | [15] |
| 9 | *Setaria adhaerens* | 1987 | Photosystem II inhibitors (5) | Atrazine | Corn | [15] |
| 10 | *Solanum nigrum* | 1987 | Photosystem II inhibitors (5) | Atrazine | Corn | [14] |
| 11 | *Conyza bonariensis* | 1987 | Photosystem II inhibitors (5) | Simazine | Orchards | [4,16] |
| 12 | *Amaranthus albus* | 1987 | Photosystem II inhibitors (5) | Simazine | Orchards | [14] |
| 13 | *Chenopodium album* | 1987 | Photosystem II inhibitors (5) | Atrazine | Corn | [16] |
| 14 | *Amaranthus cruentus* | 1989 | Photosystem II inhibitors (5) | Atrazine | Corn | [17] |
| 15 | *Bromus tectorum* | 1990 | PSII inhibitor (ureas and amides) (5) | Chlorotoluron | Wheat | [18] |
| 16 | *Bromus tectorum* | 1990 | Photosystem II inhibitors (5) | Simazine | Orchards | [19] |
| 17 | *Alopecurus myosuroides* | 1991 | PSII inhibitor (ureas and amides) (5) | Chlorotoluron | Wheat | [20] |
| 18 | *Polygonum lapathifolium* | 1991 | Photosystem II inhibitors (5) | Atrazine | Corn | [13] |
| 19 | *Alopecurus myosuroides* | 1991 | ACCase inhibitors (1), PSII inhibitor (Ureas and amides) (5) | Diclofop, chlorotoluron | Wheat | [21] |
| 20 | *Setaria verticillata* | 1992 | Photosystem II inhibitors (5) | Atrazine | Corn | [15] |
| 21 | *Echinochloa crus-galli* | 1992 | Photosystem II inhibitors (5) | Atrazine | Corn | [15] |
| 22 | *Lolium rigidum* | 1992 | ACCase inhibitors (1), PSII inhibitor (ureas and amides) (5) | Diclofop, chlorotoluron | Wheat | [22] |
| 23 | *Lolium rigidum* | 1992 | ACCase inhibitors (1) | Diclofop | Wheat | [23] |
| 24 | *Lolium rigidum* | 1992 | Photosystem II inhibitors (5) | Simazine | Orchards | [22] |
| 25 | *Echinochloa crus-galli* | 1992 | Synthetic Auxins (4) | Quinclorac | Rice | [24] |
| 26 | *Papaver rhoeas* | 1993 | ALS inhibitors (2), Synthetic Auxins (4) | Tribenuron, 2,4-D | Cereals, wheat | [25] |
| 27 | *Alisma plantago-aquatica* | 2000 | ALS inhibitors (2) | Bensulfuron | Rice | [16] |
| 28 | *Cyperus difformis* | 2000 | ALS inhibitors (2) | Bensulfuron, penoxsulam | Rice | [26] |
| 29 | *Conyza sumatrensis* | 2001 | ALS inhibitors (2) | Imazapyr | Roadsides | [27] |
| 30 | *Aster squamatus* | 2001 | ALS inhibitors (2) | Imazapyr | Roadsides | [28] |
| 31 | *Setaria viridis* | 2002 | ACCase inhibitors (1) | Sethoxydim, haloxyfop | Onion | [29] |
| 32 | *Avena sterilis* | 2005 | ALS inhibitors (2) | Imazamethaben | Winter cereals | [8] |
| 33 | *Avena sterilis* | 2005 | ACCase inhibitors (1) | Fenoxaprop | Winter cereals | [8] |
| 34 | *Conyza bonariensis* | 2004 | EPSP synthase inhibitors (9) | Glyphosate | Orchards | [30] |
| 35 | *Conyza canadensis* | 2006 | EPSP synthase inhibitors (9) | Glyphosate | Orchards | [31] |
| 36 | *Lolium multiflorum* | 2006 | EPSP synthase inhibitors (9) | Glyphosate | Orchards | [11] |
| 37 | *Lolium rigidum* | 2006 | EPSP synthase inhibitors (9) | Glyphosate | Orchards | [11] |
| 38 | *Sinapis alba* | 2007 | ALS inhibitors (2) | Tribenuron, iodosulfuron | Winter wheat | [32] |
| 39 | *Conyza sumatrensis* | 2009 | EPSP synthase inhibitors (9) | Glyphosate | Orchards | [33] |
| 40 | *Leptochloa fascicularis* | 2010 | ALS inhibitors (2) | Penoxsulam | Rice | [34,35] |
| 41 | *Leptochloa fascicularis* | 2010 | ACCase inhibitors (1) | Profoxydim, cyhalofop | Rice | [34,35] |
| 42 | *Leptochloa fascicularis* | 2010 | ALS inhibitors (2), ACCase inhibitors (1) | Profoxydim, cyhalofop, penoxsulam | Rice | [34,35] |
| 43 | *Leptochloa uninervia* | 2010 | ALS inhibitors (2) | Penoxsulam | Rice | [34,35] |
| 44 | *Leptochloa uninervia* | 2010 | ACCase inhibitors (1) | Cyhalofop | Rice | [34,35] |
| 45 | *Leptochloa uninervia* | 2010 | ALS inhibitors (2), ACCase inhibitors (1) | Profoxydim, cyhalofop, penoxsulam | Rice | [34,35] |
| 46 | *Echinochloa crus-galli* | 2010 | ALS inhibitors (2) | Penoxsulam | Rice | [36] |
| 47 | *Echinochloa crus-galli* | 2010 | ACCase inhibitors (1) | Profoxydim, cyhalofop | Rice | [36] |
| 48 | *Sinapis arvensis* | 2011 | ALS inhibitors (2) | Tribenuron, iodosulfuron | Cereals | [4] |

**Table 2.** *Cont.*

| Number | Species | Year | MoAs | Herbicides | Crops/Situations | References |
|:---:|:---:|:---:|:---:|:---:|:---:|:---:|
| 49 | *Lolium rigidum* | 2013 | ALS inhibitors (2), ACCase inhibitors (1), PSII inhibitor (ureas and amides) (5) | Chlortoluron, chlorsulfuron, diclofop | Cereals | [37] |
| 50 | *Sorghum halepense* | 2015 | ALS inhibitors (2) | Nicosulfuron | Corn | [38] |
| 51 | *Alopecurus myosuroides* | 2015 | ACCase inhibitors (1), ALS inhibitors (2), PSII inhibitor (ureas and amides) (5) | Fops, dens, cloransulam, ureas, SUs | Wheat, canola, peas, winter barley, faba beans | [4] |
| 52 | *Echinochloa crus-galli* | 2015 | ALS inhibitors (2) | Nicosulfuron | Corn | [4,39] |
| 53 | *Sorghum halepense* | 2015 | EPSP synthase inhibitors (9) | Glyphosate | Railways, freeways | [40] |
| 54 | *Lolium rigidum* | 2015 | ALS inhibitors (2), ACCase inhibitors (1), PSII inhibitor (Ureas, amides) (5), Very long-chain fatty acid synthesis inhibitors (15) | fops, dims, pyroxsulam, mesosulfuron, chlortoluron, prosulfocarb | Winter cereals | [41] |
| 55 | *Lolium rigidum* | 2016 | EPSP synthase inhibitors (9), PPO inhibitors (14) | Oxyfluorfen, glyphosate | Olive | [11] |
| 56 | *Amaranthus palmeri* | 2016 | ALS inhibitors (2) | Nicosulfuron | Corn, roadsides | [42] |
| 57 | *Conyza bonariensis* | 2016 | EPSP synthase inhibitors (9) | Glyphosate | Railways | [43] |
| 58 | *Conyza canadensis* | 2016 | EPSP synthase inhibitors (9) | Glyphosate | Railways | [43] |
| 59 | *Conyza canadensis* | 2017 | ALS inhibitors (2), EPSP synthase inhibitors (9) | Tribenuron, glyphosate | Olive | [44] |
| 60 | *Echinochloa colona* | 2017 | ALS inhibitors (2) | Penoxsulam | Rice | [36] |
| 61 | *Rapistrum rugosum* | 2018 | ALS inhibitors (2) | Tribenuron, iodosulfuron | Winter cereals | [45] |
| 62 | *Hordeum murinum* | 2018 | EPSP synthase inhibitors (9) | Glyphosate | Orchards, olive | [46] |
| 63 | *Echinochloa crus-galli* | 2018 | EPSP synthase inhibitors (9) | Glyphosate | Olive, citrus, orchards, corn, rice | [12] |
| 64 | *Bromus madritensis* | 2018 | EPSP synthase inhibitors (9) | Glyphosate | Winter cereals | [4] |
| 65 | *Bromus rubens* | 2018 | EPSP synthase inhibitors (9) | Glyphosate | Orchards, olive | [47] |
| 66 | *Conyza bonariensis* | 2019 | ALS inhibitors (2), Synthetic Auxins (4), EPSP synthase inhibitors (9), PDS inhibitors (12), PSI electron diverters (22) | Tribenuron, 2,4-D, glyphosate, diflufenican, paraquat | Olive | [48] |

A previous review on herbicide resistance in the Iberian Peninsula was conducted by Calha and co-authors in 2008 [8]. Compared with the previous report, there are now (2022) 33 species with 77 unique HR cases in this geographic area (Figure 1). Two decades ago, 26 species evolved herbicide resistance with 31 unique cases. Therefore, after around two decades, there has been a greater than two-fold increment, representing a 250% boost in HR cases in Spain and Portugal.

The weed genera with the most HR cases are *Conyza* (8), *Amaranthus* (7), and *Lolium* (5) (Figure 1). The main modes of action (MoAs) of herbicides with resistance are the ALS (acetolactate synthase), ACCase (acetyl-CoA carboxylase), EPSPS (5-enolpyruvylshikimate-3-phosphate), and PSII (photosystem II) inhibitors, with 30, 9, 16, and 7 cases, respectively (Figure 2). Additionally, the number of cases with multiple-HR, that is, with resistance to at least two different MoAs, is already 11 (Figure 2, green line). In Figure 1, it can be observed that there has been a remarkable increase in the number of cases, as well as multiple-HR cases, in the last 10 years.

The objectives of this review are to describe the current situation of resistance to herbicides in the Iberian Peninsula (Spain and Portugal), compile and summarize all known resistance cases, and discuss them in terms of the main MoAs with HR cases (ACCase, ALS, EPSPS and PSII-inhibiting herbicides), and also in terms of the most problematic HR weed species: *Lolium* spp., *Echinochloa* spp., *Papaver rhoeas*, *Conyza* spp. and *Amaranthus palmeri*. Particular emphasis will also be given to multiple-herbicide resistance and resistance conferred by enhanced rates of herbicide metabolism, due to the threat that both pose to sustaining food production [3]. The MoAs are not described in detail in this review, but a summary of their molecular targets is provided (Table 3). There are 26 MoAs recognized by the Herbicide Resistance Action Committee (HRAC), an industry

organization that monitors herbicide resistance [49]. The new universal nomenclature for MoA Classification—which uses numbers instead of letters for each one and has been in agreement with the HRAC and the Weed Science Society of America (WSSA) since 2020—will be followed, since it will prevail worldwide in the future.

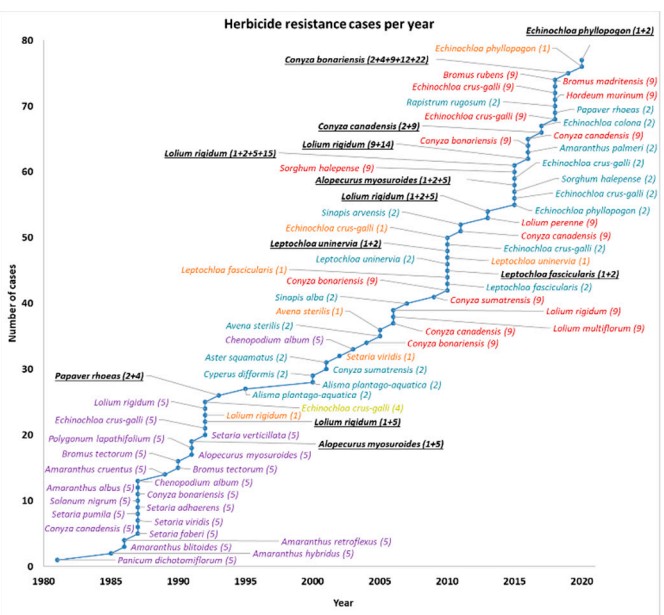

**Figure 1.** History of reports of herbicide-resistant weeds in the Iberian Peninsula, Spain, and Portugal. Chart was constructed based on the information available in the International Survey of Herbicide-resistant Weeds database [4], plus the additional cases reported in this review (Tables 1 and 2). Each color in each species represents one different mode of action (number in parentheses, according to new HRAC-WSSA classification); multiple-herbicide-resistant species (≥two modes of action) in black color and underlined.

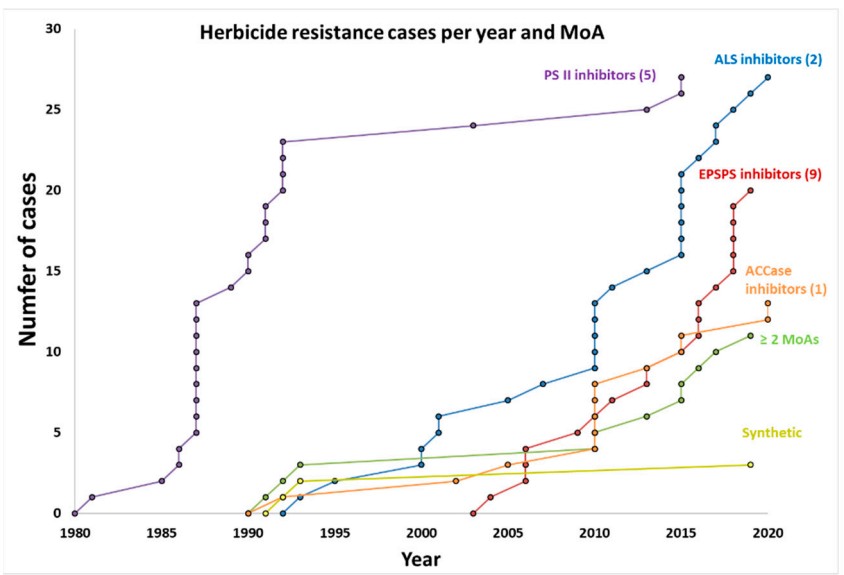

**Figure 2.** History of reports of herbicide-resistant weeds in the Iberian Peninsula per mode of action (MoA) of herbicide (different colors, number in parenthesis, according to new HRAC-WSSA classification), plus history of reports of multiple-herbicide-resistant weeds in the Iberian Peninsula to ≥two modes of action (green color).

**Table 3.** Molecular targets of the 26 herbicide modes of action recognized by the HRAC and WSSA, together with the new classification system based on numbers (first column) and the former based on letters (second column). Adapted from the HRAC [49].

| HRAC & WSSA | Legacy HRAC | Molecular Target |
|---|---|---|
| 1 | A | Inhibition of Acetyl-CoA carboxylase (ACCase) |
| 2 | B | Inhibition of Acetolactate synthase (ALS) |
| 3 | K1 | Inhibition of microtubule assembly |
| 4 | O | Auxin mimics |
| 5 | C1, C2 | Inhibition of photosynthesis. Photosystem II (PSII)—Serine 264 binder |
| 6 | C3 | Inhibition of photosynthesis. PSII—Histidine 215 binder |
| 9 | G | Inhibition of 5-enolpyruvylshikimate-3-phosphate synthase (EPSPS) |
| 10 | H | Inhibition of glutamine synthetase |
| 12 | F1 | Inhibition of phytoene desaturase (PDS) |
| 13 | F4 | Inhibition of 1-deoxy-D-xylulose 5-phosphate synthase (DOXP) synthase |
| 14 | E | Inhibition of protoporphyrinogen oxidase (PPO) |
| 15 | K3 | Inhibition of very long-chain fatty acid (VLCFA) synthesis |
| 18 | I | Dihydropteroate (DHP) synthase inhibition |
| 19 | P | Auxin transport inhibitors |
| 22 | D | Photosystem I (PSI) electron diversion |
| 23 | K2 | Inhibition of microtubule organization |
| 24 | M | Uncouplers |
| 27 | F2 | Inhibition of 4-Hydroxyphenylpyruvate dioxygenase (HPPD) |
| 29 | L | Inhibition of cellulose synthesis |
| 30 | Q | Inhibition of fatty acid thioesterase |
| 31 | R | Inhibition of serine threonine protein phosphatase |
| 32 | S | Inhibition of solanesyl diphosphate synthase |
| 33 | T | Inhibition of homogentisate solanesyltransferase |
| 34 | F3 | Inhibition of lycopene cyclase |
| 0 | Z | Unknown mode of action |

## 2. Resistance by Modes of Action

### 2.1. Resistance to ACCase Inhibitors (Group 1)

Eight weed species have been reported with resistance to ACCase inhibitors in Spain and Portugal, representing fifteen unique HR cases, with seven of them having multiple resistance to other MoAs (Tables 1 and 2, Figure 1). *Alopecurus myosuroides* (1991) was the first species identified with multiple-herbicide resistance in wheat toward diclofop-methyl and chlortoluron in Spain [21]. Afterwards, *L. rigidum* in 1992 [22]; *Setaria viridis* in 2002 [29]; *Avena sterilis* in 2005 [5]; *Echinochloa* spp. in 2010 [36]; *Leptochloa fascicularis* and *L. uninervia* in 2010 [34,35]; and *E. phyllopogon* in 2020 (Calha, personal communication) were identified with HR populations mainly in wheat and rice fields, but also in other arable crops, and also with multiple-HR biotypes (Tables 1 and 2).

Most of the reported cases were found in Spain, while there were no registered cases of lack of efficacy in winter cereals and rice in Portugal. However, resistance to ACCase inhibitors in this country has only been confirmed recently in *E. phyllopogon* from rice (Table 1). In cereals, diclofop-methyl was not so important. The selection pressure imposed by ACCase inhibitors might not have been as intense as in Spain, to promote the evolution of resistance.

Among the mechanisms involved in this resistance to ACCase-inhibitor herbicides, TSR due to point mutations is most frequent in *L. rigidum* [41], *E. crus-galli* [36] or *Leptochloa* spp. [34,35]. On the other hand, resistance in *A. myosuroides* was probably not due to an altered target site [50], while enhanced levels of herbicide detoxification have been found in some *L. rigidum* biotypes, one without mutations detected [34]. In addition, previous research showed that reduced absorption could aid in conferring resistance to ACCase inhibitors in *L. rigidum* from Spain [51].

*2.2. Resistance to ALS Inhibitors (Group 2)*

The first case of resistance to ALS inhibitors was *Papaver rhoeas*, reported in 1993 in cereal fields in the region of Catalonia in north-eastern Spain [52,53], followed by *Alisma plantago-aquatica* in 1995 from rice fields in Portugal [9]. Then, resistant biotypes of *Cyperus difformis* in 2000 [26,54], *C. sumatrensis* [27] and *Aster squamatus* in 2001 [28]; *A. sterilis* in 2005 [5]; *Sinapis alba* in 2007 [32]; and *S. arvensis* in 2011 [4] were described in different regions of Spain and Portugal. Other specific, but not minor, cases of resistance to ALS inhibitors have been confirmed more recently, such as *A. palmeri* in 2016 [42], *C. canadensis* in 2017 [44], *E. colona* in 2017 [36], *Rapistrum rugosum* in 2018 [45], and *C. bonariensis* in 2019 [48]. In total, resistance to this MoA has been detected in 20 different weed species (Tables 1 and 2).

Similarly to ACCase inhibitors, due to an altered target protein, TSR is the most reported mechanism of resistance. Mutations in positions Ala122, Pro197, and Trp574 have been detected in different grass and broadleaf weeds, while alterations in position Ser653 in the new invasive species *A. palmeri*, together with *Echinochloa* spp., have been confirmed [34,42]. For example, up to three different amino acid substitutions have been found in *C. difformis* from different rice-producing areas of the region [34]. Furthermore, evidence of the presence of NTSR, obtained by means of enhanced metabolism to ALS inhibitors, have been found in some species such as *P. rhoeas* [53,55], *L. rigidum* [41], and *C. bonariensis* [48]. In other species, the fact that no mutations were found in the ALS gene of some resistant populations points to the occurrence of NTSR mechanisms too.

*2.3. Resistance to PSII Inhibitors (Groups 5 and 6)*

With the new classification agreed upon between HRAC and WSSA, the former group C is now divided into two MoAs—group 5 and group 6 [49]—defined by the amino acid position to which the herbicide binds in the subunit D1 to inhibit PSII functioning: serine 264 for group 5, and histidine 215 for group 6 (Table 3). The new MoA 5 includes triazines (former C1) and ureas (former C2), for which resistance has been reported in Spain and Portugal. Nowadays, only terbuthylazine among triazines remains registered in Europe, mainly for corn; however, since 2020, its use has been restricted to a single application every three years. Regarding ureas, only chlortoluron is still marketed in winter cereals, and metribuzine among triazinones.

Resistance to triazines represents the first reported cases reported in the geography covered by this review (Table 2). Several grass and broadleaf weeds developed resistance to this group early in 1990s, mainly in corn, vineyards, and orchards, where atrazine and simazine were massively applied [32–34]. In *Amaranthus* species, cross-resistance between triazines and ureas within group 5 of the PSII inhibitors was reported [56]. Additionally, in *Amaranthus* triazine-resistant biotypes, the typical fitness cost associated has been described, as for *S. nigrum* [57,58]. In *Chenopodium album*, there were cases of atrazine resistance in Portugal [59]. However, no studies have attempted to determine the resistance mechanisms to this group.

Reports on resistance to the substituted ureas also go back to the 1990s for some grass weed species—*Bromus tectorum* [18], *Alopecurus myosuroides* [20] and *L. rigidum* [22]—all to chlortoluron in wheat fields. Nowadays, biotypes of multiple-HR to group 5 and other MOAs are becoming widespread in Spain (Section 3). Research to unravel the resistance mechanisms involved points to enhanced degradation, endowed by the cytochrome P450 monooxygenase (P450) family in the three species [18,20,22]. However, it is important to stress that this resistance could already have conferred or predisposed some grass weeds to cross-resistance to other MoAs marketed afterwards for their control.

Among group 6, nitriles are not available anymore in Europe, while bentazone and pyridate remain registered. No resistance cases are known to group 6 in the Iberian Peninsula. Though complaints on HR *P. rhoeas* control with bromoxynil mixtures were reported, putative multiple-HR to this herbicide has not been fully demonstrated to date.

Control failures with this contact herbicide were attributed to late timings, advanced growth stages, and high densities (Torra, unpublished).

### 2.4. Resistance to EPSPS Inhibitors (Group 9)

Glyphosate (HRAC legacy group G) is the most used herbicide worldwide [60], and represents the sole non-selective post-emergence herbicide registered in Europe nowadays. Until 2004, there were no reported resistance cases of this EPSPS inhibitor in the Iberian Peninsula, when the first case of *C. bonariensis* was identified in Spain [30]. Unfortunately, since then, there has been a steady and rapid increase in the number of reported cases, of which there are up to 18; these include eight grass weed species and the three most typical *Conyza* species, *C. bonariensis*, *C. canadensis* and *C. sumatrensis* (Tables 1 and 2), which are fully covered in Section 4.3.

Resistance to glyphosate mainly affects several perennial crops, but also non-crop lands, corn, and rice (Tables 1 and 2). To manage resistant populations, an increasing selection pressure with other MOAs has been implemented, rather than lowering the dependence on herbicides integrating chemical and non-chemical strategies. The sad consequence is that in the last 5 years, multiple-HR to EPSPS and protoporphyrinogen oxidase (PPO) inhibitors (group 14, past E, Table 3), and to ALS end EPSPS inhibitors has already been reported, both in olive and in orchards [11,44].

Both TSR and NTSR mechanisms were found among those glyphosate R species that were studied. For example, point mutations in the *EPSPS* gene can confer resistance in *Lolium* spp. [61]. Among the NTSR mechanisms, reduced absorption/translocation seem to contribute to resistance in most species investigated [28,46–48]. Astonishingly, there is already an *E. crus-galli* population reported to be able to metabolize glyphosate [12].

### 2.5. Resistance to Other Modes of Action

There are only five remaining resistance cases that do not involve the MoAs covered in previous sections. Besides resistance to quinclorac in *E. crus-galli* from rice [24]—an herbicide from a particular family (quinoline-carboxylates) within Synthetic Auxin Herbicides (SAH), group 4 (former group O, Table 3)—the other four cases involve multiple-HR populations. Resistance to ALS and SAH herbicides was already reported in *P. rhoeas* in 1998 [25]. Since then, in the last decade, it has been confirmed to be resistant to VLCFA synthesis [41] (group 15, Table 3) and PPO [11] inhibitors in *L. rigidum* populations from winter cereals and olive orchards, respectively; moreover, it is resistant to diflufenican, paraquat, and 2,4-D (groups 12, 22 and 4 in Table 3, respectively) in *C. bonariensis* from olive orchards, too [48].

NTSR mechanisms seem to drive resistance to all these particular MOAs. Besides reduced paraquat translocation in *C. bonariensis*, the common denominator between these resistance cases is the presence of enhanced metabolism [41,48,62]. There are already two broadleaf weed species with resistance to SAH in the Iberian Peninsula, which is noticeable because it is acknowledged as rare in Europe, particularly without transgenic crops [63].

## 3. Multiple-Herbicide-Resistant Weeds

As mentioned above, in the case of glyphosate, the response to resistance to one MoA in one weed species has been to increase selection pressures with another MoA, rather than integrating non-chemical options to reduce over-reliance on herbicides. Therefore, in the last decade, there has been a shift in the number of multiple-HR populations (Figure 2), some of them already reported in previous sections. Multiple resistance was already reported early in the 1990s, with an *L. rigidum* population resistant to chlortoluron and with multiple resistance to diclofop [16]; an *A. myosuroides* population with the same resistance profile [21]; and *P. rhoeas* with resistance to tribenuron and 2,4-D [52], all from wheat fields (Table 2). Today, there are nine cases of multiple-herbicide resistance reported in Spain. The most common resistance profile is multiple resistance to ACCase and ALS inhibitors in grass weeds from rice or winter cereals (Table 2). Additionally, there are already three cases

of multiple-herbicide resistance involving glyphosate. It is not surprising that the three MoAs with most resistance cases worldwide—ACCase, ALS and EPSPS inhibitors [4]—are those more present in the region with multiple resistance.

Since the response to the resistance to two MoAs has been the overuse of a third one again, multiple resistance to three MOAs has already been found, particularly in *L. rigidum*. A population from olive orchards is multiple-resistant to EPSPS, glutamine synthase, and PPO inhibitors [11], while populations with evolved resistance to ACCase, ALS and PS II are common nowadays in winter cereals [37], and some of them also, already, to VLCFA synthesis inhibitors [41]. The newest and most striking case is a *C. bonariensis* biotype with multiple resistance to five MoAs [48]. Lastly, researchers hypothesized that two main evolutionary processes explain the evolution of multiple resistance: the accumulation/stack of TSR mechanisms to different MOAs, and the evolution of enhanced metabolism which confers multiple- and cross-resistance to several MOAs.

Enhanced metabolism is the most threatening NTSR that can evolve in weeds [3]. This resistance mechanism can confer an increased activity of enzymes involved in herbicide metabolism; this is particularly concerning because it may confer multiple- and cross-resistance to herbicides with different chemical properties. Importantly, it has been demonstrated that enhanced metabolism can confer resistance to herbicides that have never been applied or marketed before [64]. Therefore, it is critical to fully understand the molecular, biochemical, and physiological mechanisms that lead to enhanced metabolism in HR weeds.

In the 1990s, the presence of enhanced metabolism as a resistance mechanism to PS II inhibitors was already inferred in *A. myosuroides* from winter cereals [65]. Among the studies trying to decipher the enzymes responsible for this enhanced degradation capacity in the region covered here, the most reported enzyme family is P450. On the other hand, the latest studies have also demonstrated the presence of higher quantities of a gluthatione-S-transferase in some *L. rigidum* biotypes from winter cereals with multiple resistance [41]. Finally, it is worth highlighting that an *E. crus-galli* biotype with enhanced capacities for degrading glyphosate has already been described in the region [12]. The first enzyme able to degrade this EPSPS inhibitor, an aldo–keto reductase in a resistant biotype, was demonstrated for the first time worldwide in *E. colona* [66].

## 4. Main Herbicide-Resistant Weed Species

Based on the search criteria specified in previous sections, the main resistant weeds have been selected according to the following criteria: permanence over time, abundance, crops affected, and particularly, the number of HR cases reported (Tables 2 and 3, Figure 1) or the presence of multiple resistance. All these species are considered among the top 15 worst HR weeds across the globe [4], excepting *P. rhoeas*, the importance of which is restricted to the Mediterranean area. Special emphasis is given to the new invasive weed *A. palmeri*, recently reported in Spain with HR biotypes [42], due to the menace it represents for summer crops.

### 4.1. *Lolium* spp.

*Lolium rigidum* (Gaud.) (rigid ryegrass), among the most crop-damaging weeds in the world, is a cross-pollinated, genetically diverse, and globally distributed species that is able to evolve resistance to multiple MoAs [4]. In Spain and Portugal, resistance to ACCase, ALS, PSII, EPSPS, glutamine synthase, VLCFA synthesis, and PPO-inhibiting herbicides has been reported in *L. rigidum*, which is mainly associated with winter cereals, vineyards and orchards [4,11,37,41,52,61,67].

In perennial crops, oxyfluorfen, alone or mixed with glyphosate and glufosinate, has been introduced as a chemical option to control dicot and grass weeds, but also to prevent and/or control glyphosate-resistant *Lolium* spp. populations [11,61]. In Portugal, glyphosate-resistant *L. rigidum* and *L. perenne* populations were identified in the Douro vineyards (north), and the alternative herbicide most used was quizalofop-P-buthyl [68].

In winter cereals, in response to multiple- and cross-resistance to ACCase, ALS and/or PSII inhibitors in *L. rigidum*, there was an increase in the use of alternative MoAs in pre-emergence, such as VLCFA synthesis inhibitors [37,41]. Therefore, the lack of an integrated weed management (IWM) approach, including non-chemical options, to manage resistance has prompted the evolution of multiple- or cross-resistant populations to the mentioned MoA in these crops, becoming more widespread in the Iberian Peninsula. Currently, *L. rigidum* is the most important and worrying HR weed in the region [69].

As expected, a variety of TSR and NTSR resistance mechanisms to different herbicides have evolved in Iberian field populationns. Point mutations have been found in the ACCase, ALS, and EPSPS genes of resistant plants [12,41,70,71]. Among the NTSR mechanisms, reduced absorption/translocation has been described for ACCase inhibitors [51] and glyphosate [61], while enhanced metabolism seems to be involved in the resistance to ACCase, ALS, PSII and/or VLCFA synthesis inhibitors [41]. Finally, fitness costs may be associated with resistance to post-emergence herbicides in winter cereals in Spanish biotypes [72].

### 4.2. *Echinochloa* spp.

*Echinochloa crus-galli* L. (barnyard grass) has evolved herbicide resistance mostly in two crops: rice and corn. The first resistance cases in rice were reported to propanil (PS II inhibitor) and quinclorac (SAH) in the 1990s [16]. Since then, resistance to post-emergence herbicides (ACCase and ALS inhibitors) has spread in all the rice-producing areas: Andalusia, the Ebro Basin, Extremadura, Navarra, Portugal and Valencia. Apart from rice monocrop, the restricted number of available MoAs (mostly ALS and ACCase inhibitors) in this crop has prompted the evolution of HR populations.

In corn, resistance to ALS inhibitors is present in some *E. crus-galli* populations from the Lleida province [4], mainly associated with flood irrigation; however, the affected surface is very restricted [69]. Resistance to triazines is very old (1992), and has not been relevant since they were banned in Europe [24]. Very recently, resistance to glyphosate has been found in several populations from Spain and Portugal, both in crop and non-crop lands [12]. Among crops, resistance has been reported in maize fields where glyphosate was used in pre-seeing, in one population in Andalusia and another one in Portugal.

Both in rice and in corn, the most frequent point mutations conferring resistance to ALS inhibitors in *Echinochloa* populations have been Pro197 and Trp574, while Ser653 has only been found in rice populations [36,39]. For ACCase inhibitors, the positions found to provide resistance in rice are Ile1781, Trp1999, and Trp2020 [34,36]. However, some of ALS-resistant populations do not harbor point mutations in the *ALS* gene. In this sense, the presence of an enhanced metabolism, i.e., to penoxsulam, is suspected in rice [36]. It is also suspected that there is a presence of multiple-HR populations to both MoAs, which will probably be confirmed in the near future. For glyphosate, among the NTSR mechanisms, reduced absorption and translocation were detected in several populations, and though TSR was not studied, EPSPS (basal) activities pointed to the presence of both point mutations and overexpression [12]. The most striking case was a population from Portugal that is able to metabolize half of the absorbed glyphosate compared to the susceptible populations [12].

### 4.3. *Conyza* spp.

The three most crop-damaging *Conyza* species (fleabanes) are present in Spain and Portugal: *C. bonariensis* (L.) Cronquist, *C. canadensis* (L.) Cronquist, and *C. sumatrensis* (Retz.) E. Walker. These species are prone to evolving resistance to glyphosate [4]. In the last two decades, several resistance cases have been described in the Iberian Peninsula for the three species in almond and fruit trees; for *C. bonariensis* and *C. canadensis* in railway margins, and for *C. sumatrensis* in olive trees [30,31,33,43,69]. Resistance mechanisms to glyphosate in these populations have only been investigated in *C. canadensis*, where reduced herbicide absorption and translocation were the main mechanisms described [31,44], though enhanced metabolism rates were found as a potential secondary mechanism contributing to resistance [31]. Recently, this potential mechanism has been described in *C. bonariensis*,

too [73]. In addition to NTSR mechanisms, it is known that the three *Conyza* species can evolve TSR mechanisms, both point mutations or overexpression of the target EPSPS [64].

In Spain, one *C. sumatrensis* population from roadsides evolved resistance to imazethapyr, an ALS inhibitor. Though not fully investigated, the high resistance factors estimated pointed to an alteration in the herbicide target as a resistance mechanism [27]. *Conyza* spp. can also evolve multiple-herbicide resistance and, unfortunately, it has already been found Spanish *C. canadensis* populations from olive trees with multiple resistance to ALS inhibitors and glyphosate [44]. The latest and most worrying case is a *C. bonariensis* biotype from olive HR to EPSPS, PS I, PDS, ALS inhibitors, and SAH [32]. It is important to notice that among the NTSR mechanisms investigated, P450 was involved in the resistance response to 2,4-D and tribenuron-methyl. Finally, in the 1980s, populations with resistance to triazines were already described in Spain [16]. In Portugal, *C. bonariensis* resistant to glyphosate were identified in olive groves in Alentejo (south) [74] and also in a citrus orchard [75], while *C. canadensis* populations were confirmed to be resistant in olive groves. TSR was investigated, but no known mutations were found in resistant populations [76].

In the area covered by this review, it is clear that *Conyza* species can evolve resistance to several MoAs, and also multiple-herbicide resistance to as many as five MOAs [48]. A variety of resistance mechanisms can be responsible depending on the herbicide, but the potential evolution of enhanced metabolism is very concerning. Therefore, the use of IWM programs, including both chemical and nonchemical tools, should be the approach to prevent and ameliorate the problem. The use of label rates, correct application timings (not beyond the rosette stage) and alternative MoAs (better mixed than sequential), are highly recommended, and almost mandatory.

### 4.4. Papaver rhoeas

Corn poppy (*Papaver rhoeas* L.) is the most abundant broadleaf weed in arable crops from Southern Europe [53,77]. The Iberian Peninsula harbors herbicide-resistant populations, and moreover, it is one of the few broadleaf species in the region with field-evolved multiple-herbicide resistant biotypes. Because of this, its biology, ecology and management, using both chemical and non-chemical strategies, has been extensively studied in Spain, in the mid to long-term [77–85].

This species has evolved resistance to SAH and/or ALS inhibitors in winter cereals. Resistance is located mainly in north-eastern Spain (Figure 3). However, in the last decade, resistance to ALS inhibitors has continued to spread in the area, moving to the north-western part, and already appearing in the south, in Andalusia and Portugal (Portugal, unpublished data). Historically lower herbicide selection pressures with ALS inhibitors in Portugal probably explain this delay in the appearance of resistance, while 2,4-D is not often used [5]. In northern Spain, less frequent crop rotation between winter and summer crops compared to the south, where two-year wheat–sunflower rotation is usual, might explain the differential selection pressures and concomitant resistance evolution paces [69].

Although multiple-resistant populations to 2,4-D and tribenuron were already found early in the 1990s [52], the mechanisms were later investigated. TSR to tribenuron by means of the amino acid substitution Pro197Ser was reported from Catalan populations [25]. Later on, six different amino acid substitutions in Pro197 were found in some populations from north-eastern Spain, while among different studies, mutations in position Trp574 have never been found in Spanish biotypes [53,84]. In addition, imazamox-enhanced degradation rates were found for the first time in the species in Spain [51]. Prior studies confirmed, in several populations, that P450 was involved in ALS inhibitor degradation [55]. Therefore, both point mutations and enhanced metabolism can contribute to resistance to group 2 in *P. rhoeas*—explaining cross-resistance to different ALS-inhibiting chemistries—and can co-exist in the same plant [53].

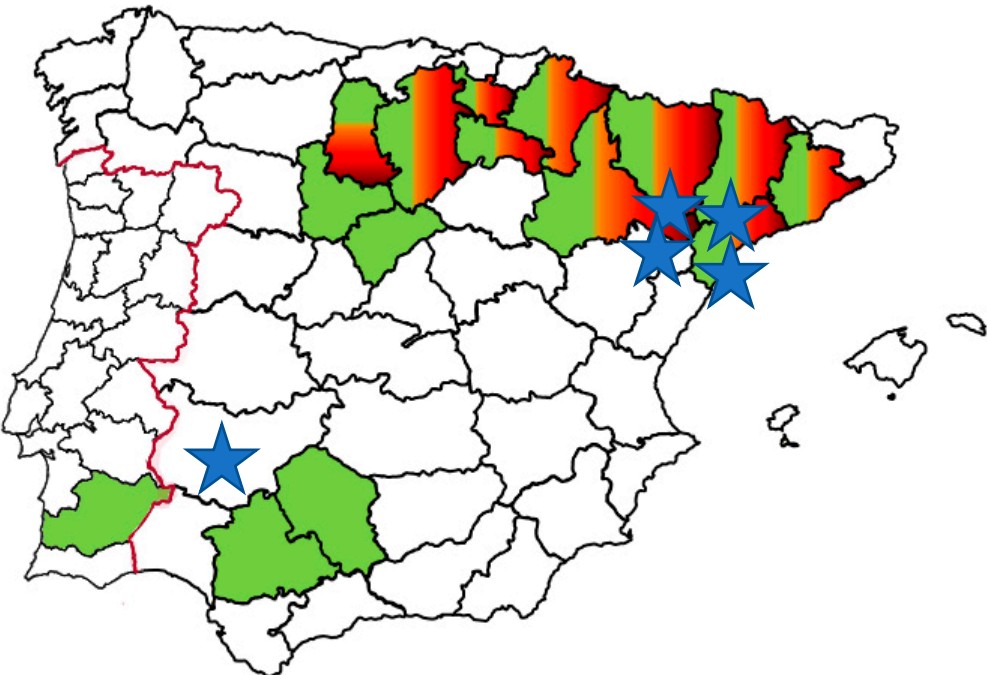

**Figure 3.** Map of resistance occurrence (provinces) for *Papaver rhoeas* (painted) and *Amaranthus palmeri* (stars). For *P. rhoeas*, green indicates resistance to ALS inhibitors, red to 2,4-D (none), and green to red indicates multiple-herbicide resistance to both. Red line denotes border between Spain and Portugal.

The first attempt to understand the basis of resistance to 2,4-D in *P. rhoeas* was in 2016 [52], where reduced translocation was described in two populations. Posterior studies found enhanced P450-based metabolism in the same, and also other, populations [55,62,86], indicating that this might be the primary resistance mechanism. Very recently, similar inhibition patterns between 2,4-D and imazamox metabolism in several resistant *P. rhoeas* populations, using different P450 inhibitors, indicated that the same P450 would be responsible of their degradation, which is concerning [55]. More studies are underway to unravel the resistance mechanisms to SAH in this species.

### 4.5. Amaranthus palmeri

*Amaranthus palmeri* S. Watson (Palmer amaranth) has very rapidly become one of the most troublesome invasive weed species in the world. Its biological attributes—mainly high fecundity, competitive ability and adaptability, and proneness to evolve multiple-herbicide resistance—explain why summer row crops are under threat [87]. This species was first detected in Spain in 2007 in the provinces of Lleida (Catalonia) and Huesca (Aragon) (Figure 3), on roadsides and field margins, where it remained [42]. Since 2016, it has entered and spread in several maize fields. Moreover, it has been confirmed that these populations harbor resistant biotypes to ALS-inhibiting herbicides [42]. *A. palmeri* resistant to ALS inhibitors has also been reported in Italy [42].

The spreading of *A. palmeri* in Spain is ongoing due to local dispersion, but also due to new arrivals from other parts of the world, mainly from America. For example, it has also been detected in the province of Badajoz in Extremadura (Figure 3), 800 km away from the other invaded area [42]. It represents a new major threat to summer row crops in the Iberian Peninsula, because new HR biotypes could arrive from the origin—i.e., those resistant to glyphosate, SAH, or other MoAs—accelerating the evolution of multiple-herbicide resistance, and highly complicating the eradication and management of the species. It is suspected that individuals resistant to glyphosate, and maybe to other MoAs, have already arrived in Spain. Studies are underway to determine their presence and putative resistance mechanisms.

### 5. Future Trends and Challenges

The data presented here should help to raise awareness on herbicide resistance and the threat it represents for Iberian agriculture, highlighting the need for proactive measures to be taken. According to this review, a total of 77 HR cases are reported in the Iberian Peninsula, with 66 in Spain and 11 in Portugal. Compared to the 44 and 6 cases found in Heap's database [4], they represent a 50% and 183% increase, respectively. For the most affected MoAs (EPSPS, ALS, and ACCAse inhibitors) or HR cases to more than two MoAs, at least a two-fold increase has occurred in the last decade (Figure 2). The scenario is particularly worrying in Spain, which now ranks first in Europe in terms of HR cases.

Herbicide resistance continues to evolve and spread in a wide range of crop weeds and management situations, both in Spain and Portugal. Therefore, a change in the trends presented in this review is not foreseen (Figures 1 and 2). It is expected that herbicide resistance will continue to steadily increase in the following years in the Iberian Peninsula. Moreover, it is certain that there are unreported herbicide-resistance cases yet to be studied in the area covered by this review. Currently, multiple-resistance to different ALS and ACCase-inhibiting herbicides is being confirmed, and mechanisms are being investigated in *Avena* spp. from arable crops, or *Echinochloa* spp. and *Leptochloa* spp. from rice. Similarly, greenhouse experiments are underway to confirm the first biotype of *Eleusine indica* resistant to glyphosate from Spain (Torra, personal communication). It is also expected that more resistance cases will be found in winter cereals in Portugal, particularly multiple-HR biotypes in *P. rhoeas* and *L. rigidum*. Finally, new HR *A. palmeri* biotype can arrive if proper quarantine measures are not undertaken.

Several factors contribute to this steady increase in herbicide resistance in the main crops in the Iberian Peninsula. Scarce new MoAs are expected, and the withdrawal or enforced dose-rate reduction of many herbicides in Europe, due to the re-registration process, will continue [88]. An important number of residual herbicides have been banned, or have been at major risk, in the last two decades; this has usually represented alternative MoAs to those to which weeds evolve resistance. The latest example is the restricted use of terbuthylazine, which has been approved since 2021 [89]. The increase in no-till cropping systems, with over-reliance on chemical control, is promoting species germinating from the soil surface, such as grasses (i.e., *Lolium* spp. or *A. myosuroides*) [90], which are prone to developing resistance. Finally, the increasing size of the fields and monocrop due to market pressures on farmers will worsen this scenario.

The number of existing effective herbicides for weed control in Europe is decreasing over time because of pesticide regulation restrictions. Additionally, the pace of MoA discovery is not enough to prevent or slow down the evolution of HR weeds [91]. Moreover, new residual herbicides such as pyroxasulfone (group 15, Table 3) will not be registered in Europe. Some new MOAs are expected to arrive, particularly in rice [92], and the first HPPD inhibitor (group 27, Table 3) will also be registered in cereals. However, a new herbicide MoA will not solve the fundamental problems of resistance in the absence of any change in the pattern of how that herbicide is used. No herbicide is invulnerable to resistance. The most striking case is glyphosate, the unique non-selective post-emergence herbicide currently registered in Europe and which will potentially be banned in the coming years.

On the other hand, the key point in slowing down selection pressure is to reduce over-all herbicide use through IWM, and a clear reliance on no-chemical and cultural strategies.

It is important to highlight that if no-chemical methods are used just to compensate weed escapes due to resistance, the approach will not solve the problem; in fact, resistant weeds will continue to spread, as shown by some studies [93]. We emphasize the importance of IWM approaches to achieve depletion of the soil seedbank. These approaches include: maximizing diversity in cropping systems and management practices, avoiding the introduction of new weed seeds, and investing more in prevention, as well as monitoring population densities. New technologies such as precision agriculture or decision support systems can help to optimize herbicide inputs, and therefore, alleviate the selection pressure on weeds.

Non-genetically modified organisms (GMOs) herbicide-tolerant (HT) crops are already marketed in Europe, such as ExpressSun®, Clearfield®, DUO System® or Provisia® crop systems. If these technologies are not imbibed in IWM programs, they will definitely contribute to (multiple-) herbicide resistance [88,94]. To our knowledge pesticide companies have stewardship programs and regulations for the rational use of these technologies and to prevent the evolution of HR weeds. It is unknown if GMOs HT crops will finally be registered and marketed in Europe. If so, they should be accompanied by proper management and marketing regulations, otherwise their contribution to the spread of herbicide resistance can bring the continent to a very worrying future.

In addition, the largest cropping areas of the Iberian Peninsula are under Mediterranean climate. Recent global warming, which exceeds global rates, exacerbates environmental stresses in the Mediterranean region [95]. Therefore, it is expected that agricultural weeds will suffer increased stress levels due to climate change in the region. Some studies suggest that plant responses to abiotic/biotic stressors are connected with NTSR-based herbicide resistance mechanisms, particularly herbicide metabolism [96]. Considering that the first herbicide-responsive (metabolic) genes are being identified [3], the future scenario of global warming might accelerate the evolution of herbicide resistance by means of reduced herbicide efficacy, maybe due to stress-responsive genes linked with detoxification routes. It is unknown if this process is already ongoing in the Iberian Peninsula, and it might explain the steady increase shown in the reporting of HR weeds in the last decade.

Finally, future perspectives and research works should address the following aspects: (1) further investigation of the relationship between management strategies and their influence on the occurrence of TSR and NTSR resistance, in order to design effective management strategies; (2) the development of effective monitoring and diagnostics tools to rapidly identify emerging herbicide resistance; and (3) deepening the study of NTSR-type resistances, specially the identification of the genes involved.

**Author Contributions:** Conceptualization, J.T.; methodology, J.T.; validation, J.T., J.M.M. and R.d.P.; formal analysis, J.T.; investigation, J.T.; resources, J.T.; data curation, J.T.; writing—original draft preparation, J.T.; writing—review and editing, J.T., J.M.M., I.M.C., M.D.O., J.P. and R.d.P.; visualization, J.T.; supervision, J.T. and R.d.P.; funding acquisition, J.T. All authors have read and agreed to the published version of the manuscript.

**Funding:** This research received no external funding.

**Institutional Review Board Statement:** Not applicable.

**Informed Consent Statement:** Not applicable.

**Acknowledgments:** Joel Torra acknowledges support from the Spanish Ministry of Science, Innovation, and Universities (grant Ramon y Cajal RYC2018-023866-I).

**Conflicts of Interest:** The authors declare no competing financial interest.

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
