# Peer review of "Current Status of Herbicide Resistance in the Iberian Peninsula: Future Trends and Challenges"

_agronomy, doi:10.3390/agronomy12040929_

Round 1
Reviewer 1 Report
Well written and well presented. Minor comments given directly in the manuscript.

Author Response
Reviewer 1
According to reviewers’ comments, the following minor revisions were addressed:
L36 changed to south-western, thanks to notice.
L43 evolution changed by development.
L151 one ‘with same’ deleted, thanks to notice.
Reviewer 2 Report
The authors have presented a review manuscript, which evaluated the status of herbicide resistance in the Iberian Peninsula: future trends and challenges. The manuscript presents interesting results concerning the selection and the response in the herbicide resistance, but they are some point, which need to improve. Following, I have included some comments aimed to enhance the paper:
- The authors must include mores details and related studies in the introduction, it is very short. Improve the introduction and the presentation of the objectives.
- Some keywords are like a text, simplify them.
- This work presents very interesting results. I think that the authors can improve the format of results demonstration. The authors can highlight better the importance of the results obtained.
· Figure number 1 can't be seen at all, it has to be improved · Consider extending the conclusions and adding a Future works paragraph.
Finally, the topic of this manuscript is interesting; since the selection of herbicide resistances, but authors must restructure the manuscript to improve the readability of the text and their future trends and challenges.
Reviewer 3 Report
The MS encompasses literature synthesis on herbicide resistance of prevalent weeds species in a specific geographical region (Iberian Peninsula). The subject matter is quite pertinent under changing climate scenario to ensure food security through optimal weed management and fits well within the scope and objectives of AGRONOMY. However there are few concerns/clarifications highlighted for consideration of the authors;
Generalized comments:
= To avoid the impression of a localized study, it is perhaps better to add information on evolving herbicide resistance in global perspectives of the same weeds species present in Iberian Peninsula.
=The write-up style seems to be of report writing instead of scientific article that needs thorough amendment by omitting too generalized statements.
=I could not trace out methodology followed by authors to synthesize the literature on the subject matter that may confuse readers, it is perhaps better to appropriately elucidate brief information on literature synthesis criteria for selection of peer-findings used during the studies stated at the end of introduction must be shifted under separate heading for reader’s convenience.
=Language of the MS needs thorough revision in order to impart clarity to the content.
Specific comments
ABSTRACT:
--The starting phrases are the most important ones to portrait the problem statement explicitly and signify the pertinence of serious threat of herbicide resistance, thus it is better to replace the starting phrase.
-- “unique herbicide resistant cases in this geographic area” in what sense unique??????
-- “Diversity of resistance mechanisms” needs further elaboration to give readers explicit idea of factors responsible for imparting herbicide resistance among weeds species.
KEYWORDS: Use high frequency words related to the subject matter.
INTRODUCTION
--It is too brief and generalized
--Authors must add missing information on
-what herbicide resistance is?
-What are the factors (both genetic and evolving agro-botanical traits of weeds species) responsible for herbicide resistance?
-Brief information on weeds resistance to prevalent herbicides in global perspectives must also be added?
-Perhaps authors have skipped the most pertinent part of the study i.e., mode of action of herbicides and their brief description of their classification based of MOA.
-Further information may be added on prevalent weed species and their extent of their economic losses?
==Table 1 is too lengthy and it is perhaps better to split into at least 2/3 tables country wise or more preferably weeds species type (broad or narrow leaf, indigenous or exotic species etc.) for readers convenience.
==Figure 1 presenting year-wise herbicide resistance cases need to be described in more detail along with information on weeds type and herbicide class.
==Figure 2 caption and heading do not match?
==Resistance by mode of action: Authors have merely synthesized information on number of weeds while there is dire need to briefly describe information about weeds species and their agro-botanical traits, for this, a separate table might be added to give readers explicit information on weeds showing herbicide resistance in Spain and Portugal respectively.
== “Main herbicide resistance weeds species” this section needs to be shifted after adding methodology section.
==Again, information is lacking about the extent of economic losses caused by these weeds.
== “Several factors contribute to this steady increase in herbicide resistance………” it needs further clarification by adding latest information/peer-findings.
==Future perspectives and research needs may also be highlighted based on synthesized data.
Round 2
Reviewer 3 Report
The revised version of the MS “Current status of herbicide resistance in the Iberian Peninsula: future trends and challenges” has been thoroughly revised by the authors in accordance with the suggestions forwarded previously. However following suggestions might be considered prior to acceptance of the MS.
Abstract
The starting phrase might be amended as
“The evolution of herbicide resistance in weeds has emerged as one of the most serious threats to sustainable food production systems which necessitates herbicides evaluation for determining their efficacy”.
“About 50 years ago the first case of herbicide resistance in the Iberian Peninsula was reported: Panicum dichotomiflorum resistant (R) to atrazine in Spanish maize fields” needs to be rephrased.
Add future perspectives of the study in one phrase in the end of abstract.
INTRODUCTION
“Moreover, in 2020 around 20.200 tons of herbicides were commercialized in the country, 4% superior than 2019”
“besides those already included in that database, were to add those found thanks to” needs rephrasing.
“The last review on herbicide resistance in the Iberian Peninsula was done by Calha and co-authors in 2008” needs rewording as it will limit the addition of latest peer findings.
Moreover, language editing must be done for imparting clarity to the content.
